# Proteinuria and Bilirubinuria as Potential Risk Indicators of Acute Kidney Injury during Running in Outpatient Settings

**DOI:** 10.3390/medicina56110562

**Published:** 2020-10-27

**Authors:** Daniel Rojas-Valverde, Guillermo Olcina, Braulio Sánchez-Ureña, José Pino-Ortega, Ismael Martínez-Guardado, Rafael Timón

**Affiliations:** 1Centro de Investigación y Diagnóstico en Salud y Deporte (CIDISAD), Escuela Ciencias del Movimiento Humano y Calidad de Vida (CIEMHCAVI), Universidad Nacional, Heredia 86-3000, Costa Rica; 2Grupo en Avances en el Entrenamiento Deportivo y Acondicionamiento Físico (GAEDAF), Facultad Ciencias del Deporte, Universidad de Extremadura, 10005 Cáceres, Spain; wismu04@gmail.com; 3Programa Ciencias del Ejercicio y la Salud (PROCESA), Escuela Ciencias del Movimiento Humano y Calidad de Vida (CIEMHCAVI), Universidad Nacional, Heredia 86-3000, Costa Rica; brau09@hotmail.com; 4Departmento de Actividad Física y Deporte, Facultad Ciencias del Deporte, 30720 Murcia, Spain; josepinoortega@um.es

**Keywords:** urine, biomarkers, renal health, assessment, mountain running, acute renal failure

## Abstract

*Background and objectives*: The purpose of this study was to explore which urinary markers could indicate acute kidney injury (AKI) during prolonged trail running in outpatient settings. *Materials and Methods*: Twenty-nine experienced trail runners (age 39.1 ± 8.8 years, weight 71.9 ± 11 kg, height 171.9 ± 8.3 cm) completed a 35 km event (cumulative positive ascend of 1815 m, altitude = 906 to 1178 m.a.s.l.) under a temperature of 25.52 ± 1.98 °C and humidity of 79.25 ± 7.45%). Two participant groups (AKI = 17 and No-AKI = 12) were made according to AKI diagnosis criteria based on pre- and post-race values of serum creatinine (sCr) (an increase of 1.5 times from baseline). Blood and urinalysis were performed immediately pre- and post-race. *Results*: Pre- vs. post-race differences in sCr and sBUN were found in both AKI and No-AKI groups (*p* < 0.01). Differences in post-race values were found between groups (*p* = 0.03). A total of 52% of AKI runners presented significant increases in proteinuria (*χ^2^* = 0.94, *p* = 0.01) and 47% in bilirubinuria (*χ^2^* = 0.94, *p* = 0.04). Conversely, No-AKI participants presented no significant increases in urine markers. *Conclusions*: These study’s findings may suggest the potential use of urinalysis as an accessible alternative in the outpatient setting to early identify transitional AKI until a clinical confirmation is performed.

## 1. Introduction

Acute kidney injury (AKI) is a condition defined as an injury or damage accompanied in some cases by renal dysfunction over a relatively short period [1,2,3]. This takes into account that human kidneys have a significant glomerular function reserve and dysfunction may be evident only when more than 50% of the total renal mass is compromised [4]. This sudden episode of kidney damage occurs within a few hours (<48 h), causing a build-up of waste products in the bloodstream, making it difficult for kidneys to maintain the body’s fluid balance. This abrupt kidney damage occurs in a wide range of clinical settings. It represents a relatively common but under-recognized problem in sports medicine and science, and AKI’s long-term effects on renal function are still unclear [5].

Transitional AKI has been reported as a severe condition with an increasing incidence in endurance sports such as triathlon, open water, swimming, cycling, and running. Approximately 16% to 50% of athletes participating in long-distance events fulfill AKI diagnosis criteria [5], and 97% of these participants are endurance runners [5]. Muscle damage and subsequent inflammatory responses could result from consecutive eccentric–concentric muscle contraction during endurance running and be AKI’s potential cause [5,6,7,8]. In strenuous and prolonged events, AKI etiology is considered multifactorial, and heat strain, dehydration, high metabolic, and physical load are potential enhancers of this temporal decrease in kidney function [5,9,10].

Some evidence has reported that blood markers’ changes seem to be a physiological reaction in sports, and it is overestimated in AKI [11]. Recent epidemiological and experimental studies have demonstrated a real link between AKI and chronic kidney disease (CKD) [12,13]. Repeated AKI episodes, even mild cases, may induce CKD over the long term. The potential link between consecutive AKI episodes and CKD has to be confirmed in sports due to the need for clarification around nature and precise mechanisms leading to AKI in these activities. It is known that prolonged exercising does not impact short-term kidney function, and it is transitional damage that usually requires a few days to recover normal function [5,14].

AKI is usually assessed using markers such as serum creatinine (sCr), albumin, serum blood ureic nitrogen (sBUN), cystatin-C, neutrophil gelatinase-associated lipocalin (NGAL), kidney injury molecule-1 (KIM-1), pro-inflammatory cytokines (e.g., IL-18), and liver-type fatty acid-binding protein [15,16,17,18]. This condition is also defined, classified, and stratified based on severity according to some classification systems, such as Risk, Injury, Failure, Loss, and End-Stage (RIFLE) and Acute Kidney Injury Network (AKIN) criteria [3,19]. This diagnosis criterion allows earlier identification of AKI and diagnosis, even in the absence of subsequent kidney dysfunction [4,20].

AKI is also associated with urine alterations [21], such as the high prevalence of proteinuria [11], hematuria [22], albuminuria [23], and creatinuria [24] with a return to baseline after a few days. Urine analysis is usually performed using urine samples analyzed in the laboratory, but urine dipstick readings are generally used as an accessible alternative [25,26]. In endurance sports, it has been shown that long races, compared to short races, present a higher incidence of urine alterations [11], so it is related to duration but not intensity [27]. These changes in urine characteristics are usually associated with hemodynamic adjustments as hemoconcentration and renal hypoperfusion due to dehydration during endurance sports [23].

The increase in wide-spread participation in endurance events has raised concerns around these activities’ potential implications for the participant’s kidney health [4,10]. Considering that adverse environmental conditions such as high temperature and humidity as in tropical settings increase AKI’s risk, renal function must be monitored in endurance sports such as running, mostly when it is practiced in hot regions [28,29]. Additionally, it is fundamental to have more available options to monitor kidney function during training and competition. Based on previous evidence, urinalysis could be an alternative and accessible method to identify AKI early in the field. These measurements could offer some vital information to allow action to be taken on affected athletes in the field, while waiting for subsequent confirmation analysis (e.g., medical imaging or blood tests). This kind of in-field analysis could provide objective data to boost prevention strategies in those diagnosed participants. Since AKI can be highlighted as a severe clinical problem with significant morbidity, the objective of this study was to explore which urinary markers could indicate AKI during prolonged trail running in outpatient settings.

## 2. Materials and Methods

### 2.1. Design

This was a retrospective cohort study where participants completed a 35 km event (cumulative positive ascend of 1815 m, altitude = 906 to 1178 m.a.s.l.). The event was held in Mora, San José, Costa Rica under a temperature of 25.52 ± 1.98 °C and humidity of 79.25 ± 7.45% (QUESTemp^TM^ 36, 3M, Saint Paul, MN, USA). Two participant groups (AKI = 17 and No-AKI = 12) were made according to AKI risk criteria based on pre- and post-race values of sCr (an increase of 1.5 times from baseline) [19]. Both blood and urinalysis were performed immediately pre- and post-race. Serum and urine samples were collected ~15 min before and ~15 min after the race (see Figure 1). Participant’s finish time was 4.5 ± 0.3 h.

### 2.2. Participants

Twenty-nine mountain runners (age 39 ± 9.1 years, weight 71.7 ± 10.8 kg, height 172.2 ± 8.3 cm) took part in the study. Participants were selected among >18 years old, voluntary, experienced (5.41 ± 2.79 years), trained (8.8 ± 3.4 h/week), and heat acclimatized (sleep and train in similar study’s altitude and weather) ultra-endurance runners. Participants were recruited from a single endurance event and reported no neuromuscular injuries or metabolic disturbances at least six months before the race.

The experimental protocol was approved by the Institutional Review Board of Universidad Nacional de Costa Rica (Reg. Code UNA-CECUNA-2019-P005; 17 June 2019) and Universidad de Extremadura (Reg. Code 139/2020; 25 September 2020). All participants were informed of the details of the experimental procedures and the associated risks and discomforts, as well as their benefits and rights. According to the criteria of the Declaration of Helsinki, each participant gave written informed consent regarding biomedical research involving human subjects (18th Medical Assembly, 1964, revised in 2013 in Fortaleza).

### 2.3. Materials and Procedures

#### 2.3.1. Serum Test

Blood was extracted in situ from the antecubital vein using a 5 mL blood collection sterile tube (Vacutainer^TM^, Becton, Dickinson & Company, Franklin Lakes, NJ, USA) containing a spray-coated silica particle activator. Tubes contained a gel polymer to facilitate serum separation during centrifugation (10 min at 2000× *g* relative centrifugal force) using centrifuge tubes (PLC-01, Gemmy Industrial Corp., Taipei, Taiwan). Blood samples were stored on ice in a special cooler (45QW Elite, Pelican Products^TM^, Torrance, CA, USA) until serum samples were frozen at −20 °C (~5 h after blood extraction). Sample analysis and processing were performed 24 h after data collection in an isolated and temperature-controlled laboratory using an automatic biochemical analyzer (BS-200E, Mindray, city, China) by photometry method. All procedures were performed under relevant protocols for the handling and disposal of biological materials, according to the manufacturer’s instructions for the equipment and reagents used. The analyzed variables were sCr (mg/dL) and sBUN (mg/dL).

#### 2.3.2. Urine Test

Urine samples were collected in situ in a 30 mL polypropylene sterile urine sample container (Nipro Medical Corp., Osaka, Japan). Samples were analyzed using highly sensitive and accurate dipsticks for urine screening (Combur_10_Test M, Roche, Mannheim, Germany) during distance running [25]. Urine dipsticks were examined immediately after collection by two different microbiologists simultaneously using the manufacturer’s color scale. In case of disagreement between observers, a consensus was obtained by the opinion of a third researcher. The following parameters were screened: leucocytes, erythrocytes, bilirubin, ketones, nitrites, protein, glucose, and urobilinogen. There were no reported urination problems or difficulties neither before nor after the race. Traces were considered as negatives, and those scores >1 were reported. Urine test interpretation and reporting were made as follows: >1 score was equivalent to leucocytes > 10 cells/µL, erythrocytes > 5 cells/µL, bilirubin > 1, ketones > 1, nitrites +, protein > 30 mg/dL, glucose > 50 mg/dL, and urobilinogen > 1 mg/dL.

#### 2.3.3. Urine Specific Gravity

Urine specific gravity (USG) was assessed as a hydration status marker. Urine solids were assessed, and USG was confirmed and double-checked with a digital valid [30] handheld refractometer (Palm Abbe^TM^, Misco, Solon, OH, USA). It was classified following the hydration status ranges: well-hydrated < 1.01, minimal dehydration 1.01–1.02, significant dehydration 1.02–1.03, and severe dehydration > 1.03 [31]. The refractometer was cleaned with distilled water and calibrated previously.

### 2.4. Statistical Analysis

The description of variables was reported using mean, standard deviation, and lower and upper limits. The normality of the data was confirmed using the Shapiro–Wilk test. Participants’ basic data and characteristics were compared using an independent *t*-test. Differences between AKI and No-AKI groups in blood biomarkers and USG were explored using a mixed analysis of variance, and post-hoc of Bonferroni was performed to look after specific differences. Omega squared (ω_p_^2^) was selected to qualify the magnitude of the differences as follows: <0.01 trivial; >0.01 small; >0.06 moderate, and >0.14 large [32].

McNemar’s non-parametric test was used to explore the possible change in proportion for the paired data of urinalysis. In those observed cases, the intersection frequency value was <5; the binomial test was performed. The data of urinalysis were paired by measurement moment using a 2 × 2 contingency table. Alpha was set at *p* < 0.05, and all data were analyzed and systematized using the e Statistical Package for the Social Sciences (SPSS, IBM, SPSS Statistics, v.22.0, Chicago, IL, USA).

## 3. Results

Table 1 shows the comparison between AKI and No-AKI groups based on the participants’ basic data and characteristics. There were no differences in this participant’s information by group.

There were 17 participants that met AKI criteria (sCr = 1.18 ± 0.26 pre, 1.81 ± 0.35 post, change of 53.4%). There were large differences by measurement (pre vs. post) and group (AKI vs. No-AKI) in sCr (F = 17.24, *p* < 0.01, ω_p_^2^ = 0.38 (large)) and sBUN (F = 4.1, *p* < 0.5, ω_p_^2^ = 0.1 (large)). Pre vs. post differences were found in both AKI (*p* < 0.01) and No-AKI (*p* < 0.01) groups in sCr and sBUN. Moreover, in sCr values, post-race differences between AKI and No-AKI groups were found (*p* = 0.03) but no pre-race differences were identified (*p* = 0.34) (see Figure 2a). Additionally, in sBUN values, there were no pre- or post-race differences between AKI and No-AKI groups (see Figure 2b).

Of AKI runners, 52.94% presented a significant increase in proteinuria (*χ^2^* = 0.94, *p* = 0.01) and 47.06% in bilirubinuria (*χ^2^* = 0.94, *p* = 0.04) when comparing pre- vs. post-race values. No significant increases were found in leucocyturia (17.64%, *χ^2^* = 04.96, *p* = 0.5), urobilonogenuria (17.64%, *χ^2^* = 0.23, *p* = 0.63), and hematuria (29.41%, *χ^2^* = 0.58, *p* = 0.13). No cases of nitrituria, glucosuria, or ketonuria were found (see Table 2).

Furthermore, 12 participants did not develop AKI (sCr = 1.28 ± 0.28 pre, 1.5 ± 0.3 post). Proteinuria (33.33%, *χ^2^* = 1.67, *p* = 0.25), ketonuria (16.66%, *χ^2^* = 0.28, *p* = 1), bilirubinuria (41.66%, *χ^2^* = 0.74, *p* = 0.63), urobilonogenuria (8.33%, *χ^2^* = 0.12, *p* = 1), and hematuria (33.33%, *χ^2^* = 2.59, *p* = 0.5) were found but with no significant change. No cases of leucocyturia, nitrituria, or glucosuria were presented in the No-AKI group (see Table 3).

Finally, USG as a hydration marker showed no significant interaction in AKI vs. No-AKI groups (F = 0.62, *p* = 0.44, ω_p_^2^ = 0 (trivial)). There were no pre vs. post differences (F = 3.1, *p* = 0.09; pre = 1.018 vs. post = 1.023).

## 4. Discussion

This study aimed to explore which urinary markers could indicate AKI during prolonged trail running in outpatient settings. The main findings of this study were as follows: (1) Pre- vs. post-race differences in sCr and sBUN were found in both AKI and No-AKI groups (*p* < 0.01); (2) differences in post-race values were found between groups (*p* = 0.03); (3) a total of 52% of AKI runners presented significant increases in proteinuria (*χ^2^* = 0.94, *p* = 0.01) and 47% in bilirubinuria (*χ^2^* = 0.94, *p* = 0.04); and, conversely, (4) No-AKI participants presented no significant increases in urine markers.

AKI could be caused by a series of factors such as decreased blood flow, direct kidney trauma, blockage of the urinary tract, among others [33]. In sports, precisely in endurance sports, the mechanisms are still unclear. Some factors such as heat strain, dehydration, and high metabolic and physical load may boost the risk and are primary issues. Dehydration seems to be a factor that did not influence the AKI occurrence in this specific sample, as found in the results. In strenuous exercise, high physical load during prolonged periods has an essential role in AKI development [6,34]. The relative typical rise in sCr values during endurance events could suggest a high muscle damage rate due to the release of sarcoplasmic proteins into the bloodstream. Damage and disintegration of muscle fibers are expected consequences of strenuous physical exertion. Distance running events are one of the most physically demanding sports, and the subsequent structural and functional damage could be exacerbated due to the repetitive concentric–eccentric muscle actions when running uphill and downhill as happen in endurance trail training and competitions. These efforts usually require greater impact absorption and a higher metabolic rate [6,35,36] as compared to other sports.

Protein is one of the main structural components of muscle fibers in the body. Under normal conditions, the kidney’s protein excreted in healthy adults is about 150mg per day [37]. As a consequence of muscle damage, an excess of proteins is excreted through the urine, and a condition known as proteinuria could develop. This condition is asymptomatic and associated with intense exercise, also called exercise-induced proteinuria [22], as was found in the present study.

The pathophysiological mechanisms of proteinuria can be partially explained by increasing glomerular capillary permeability to proteins and reduced protein reabsorption capacity in the renal tubules. Still, exercise-induced proteinuria is not fully understood, but it seems that the renin–angiotensin system and prostaglandins have an essential role in its development [38].

Proteinuria and bilirubinuria in endurance sports could be a consequence of a cascade of events in the kidney. In non-contact sports, catecholamines are released by the suprarenal glandules causing a redirection of blood to muscles and restricting kidney blood flow. These events lead to hypoxic nephron damage and an increase in glomerular permeability [39]. Vasoconstriction of the glomerular arteriole is also provoked by catecholamines, resulting in decreased glomerular filtration pressure and allowing excretion of some macro- and microscopic elements in urine as protein, erythrocytes, albumin, and bilirubin. Other factors contributing to exercise proteinuria could be, but are not limited to, lactate accumulation, oxidant stress, hormonal changes, and sepsis [22,25].

The increase in bilirubin found in this research could be caused by hemolysis and subsequent catabolism of hemoglobin. The proliferation of red blood cell breakdown is caused mainly by free radicals and a mechanical factor [40]. Bilirubinuria could also be related to hepatic disturbance during long-distance running [41,42]. Endurance running may cause a decline in hepatic function related to changes in the liver cells’ membrane by lipid peroxidation due to blood flow restrictions and free radicals’ release. It is known that the liver suffers a temporary decline in its function during prolonged exercises compared to running over shorter distances [42]. A condition called foot-strike hemolysis suggests that blood cells’ mechanical injury could be related to the consecutive impact during running [43].

Exercise-related proteinuria and bilirubinuria have been related to renal and hepatic dysfunctions. Both conditions could be asymptomatic, transitional, reversible, and, usually, they do not need any special care. However, endurance athletes could be particularly vulnerable to developing such conditions when exposing themselves to a high level of environmental stress, such as a hot and humid environment [10,44,45]. These conditions could boost ischemia, hypoxemia, and ATP depletion in renal tubular cells, and considering dehydration, it could be exacerbated by increased sodium reabsorption [29].

The presence of AKI cases with concomitant proteinuria and bilirubinuria may suggest the potential use of urinalysis as an accessible alternative to identify AKI cases early in the field and monitor training and competition as an outpatient setting. The screening of urine changes could represent an opportunity to identify the potential risk of AKI cases in a simple and fast manner. This result must be analyzed with caution, considering that only 47 to 52% of AKI runners presented urine changes.

Based on quantitative results, scientists could overlook the incidence and prevalence of AKI cases with concomitant urine findings. Still, at the clinical level, these results’ potential implications may lead stakeholders to deeply analyze those cases, although it could be considered relatively uncommon [5]. Finally, there is no clear link between AKI and more severe complications such as chronic kidney disease in endurance sports. Some actions must be addressed to prevent future health issues in athletes. Managing fluid intake and restoring electrolytes prior to, during, and after endurance events may contribute to the reduction of the number or lessen the severity of AKI cases. Avoiding repeated participation at endurance events without the required rest and recovery between exhaustive efforts could be protective against AKI.

These findings must be seen in the light of some limitations. Some contextual factors such as liquid intake, food consumption, and supplements during running should be controlled in future studies. Despite limited access, it may be interesting to assess some novel AKI indicators as Cyst-C, NGAL, and KIM-1 as subclinical AKI markers. It is fundamental to develop a cohort follow-up to confirm the potentiality of cumulative AKI events leading to CKD. Unfortunately, assessing blood and urine samples of a large cohort for research purposes during a long-distance event as trail running is not always feasible; future studies must include a greater sample size.

Homogeneity between groups and diagnosis criteria of AKI made it difficult to interpret why there were no differences between AKI vs. No-AKI groups, considering there was an occurrence of proteinuria in 33% of No-AKI cases as well as differences in sCr and sBUN between pre- and post-race assessments. This may suggest that, during clinical evaluation of AKI during endurance sports, patients may be analyzed individually to explore these findings’ real relevance and potential runner’s health impairment.

Additionally, it should be considered that dipstick analysis is usually used in most outpatient settings to semi-quantitatively measure the urine protein concentration but not the type or total amount. These tests are a crude estimation of urine protein concentration, so this is an initial approach for AKI that seems to correlate with AKI markers. Runners with persistent proteinuria should undergo a quantitative measurement of protein excretion using, for example, a 24-h urine specimen (urine protein/Cr).

## 5. Conclusions

Endurance trail running could lead to an increase in some blood and urine samples related to transitory AKI. This study found pre- vs. post-race differences in sCr and sBUN in both AKI and No-AKI groups, differences in post-race values between groups (*p* = 0.03), and a total of 52% of AKI runners presented significant increases in proteinuria (*χ^2^* = 0.94, *p* = 0.01) and 47% in bilirubinuria (*χ^2^* = 0.94, *p* = 0.04).

These results may suggest that AKI prevalence with concomitant proteinuria and bilirubinuria is relatively uncommon among endurance runners. Although these cases do not represent most of the runner’s condition, at clinical level care, these findings must be taken with precaution to prevent future complications. Furthermore, although there is insufficient evidence that links AKI to other future complications, these markers should be monitored during training and competition to prevent potential future damage.

## Figures and Tables

**Figure 1 medicina-56-00562-f001:**
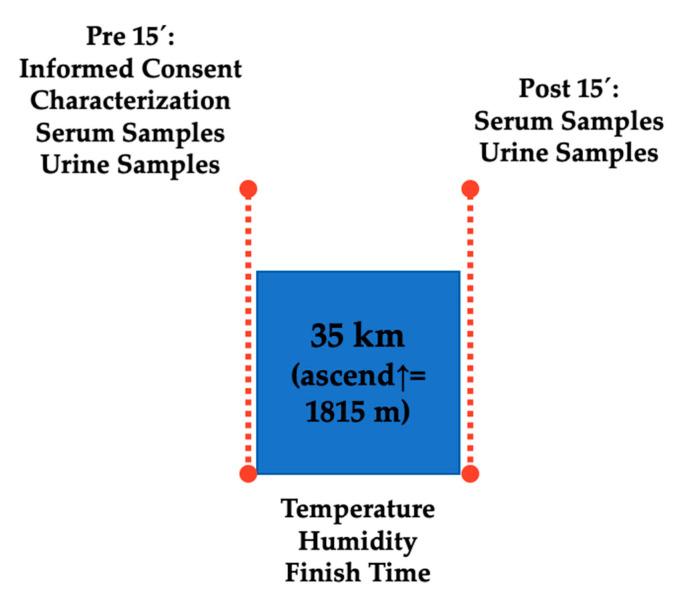
Schematic design of study assessments.

**Figure 2 medicina-56-00562-f002:**
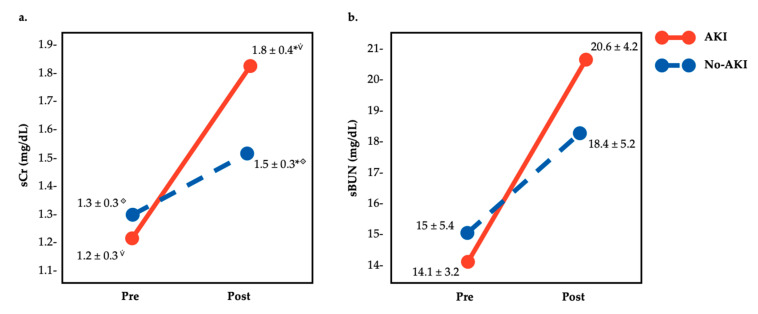
Comparison of pre- and post-race (**a**). serum creatinine (sCr), and (**b**). serum blood ureic nitrogen (sBUN) values of trail runners by the presence or not of acute kidney injury. ^⟐ ⩒^
^*^ significant statistical differences.

**Table 1 medicina-56-00562-t001:** Participants’ basic data and characteristics comparison.

	AKI	No-AKI	*t*	*p*-Value
Age (years)	39.4 ± 8.8	38.1 ± 9.8	0.34	0.74
Weight (kg)	69.2 ± 7.3	76 ± 14.5	−0.99	0.33
Height (cm)	171.6 ± 7.6	173.1 ± 9.6	0.41	0.69
Trail running experience (years)	5 ± 2.6	6.1 ± 3.1	−1.63	0.12
Training (hours)	9.1 ± 2.8	8.5 ± 4.5	−0.45	0.66

**Table 2 medicina-56-00562-t002:** Urinalysis outcomes in participants fulfilling acute kidney injury (AKI) diagnosis criteria.

Variable (Score Criteria)	Pre		Post 0 h		*χ^2^*	*p*-Value
	*n* *	%	*n* *	%		
Leucocytes (>1)	0	0	3	17.64	4.96	0.5
Nitrites (>1)	0	0	0	0	-	-
Protein (>1)	0	0	9	52.94	0.94	0.008
Glucose (>1)	0	0	0	0	-	-
Ketones (>1)	0	0	0	0	-	-
Urobilinogen (>1)	0	0	3	17.64	0.23	0.625
Bilirubin (>1)	0	0	8	47.06	0.94	0.039
Erythrocytes (>1)	0	0	5	29.41	0.58	0.125

* Based on AKI group data (*n* = 17).

**Table 3 medicina-56-00562-t003:** Urinalysis outcomes in participants without AKI diagnosis.

Variable (Score Criteria)	Pre		Post 0 h		*χ^2^*	*p*-Value
	*n **	*%*	*n **	*%*		
Leucocytes (>1)	0	0	0	0	-	-
Nitrites (>1)	0	0	0	0	-	-
Protein (>1)	0	0	4	33.33	1.667	0.25
Glucose (>1)	0	0	0	0	-	-
Ketones (>1)	0	0	2	16.66	0.278	1
Urobilinogen (>1)	0	0	1	8.33	0.123	1
Bilirubin (>1)	0	0	5	41.66	0.741	0.063
Erythrocytes (>1)	0	0	3	33.33	2.59	0.5

* Based on No-AKI group data (*n* = 12).

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
