# Peer review of "Proteinuria and Bilirubinuria as Potential Risk Indicators of Acute Kidney Injury during Running in Outpatient Settings"

_medicina, 2020, doi:10.3390/medicina56110562_

Round 1

Reviewer 1 Report

I have carefully reviewed the manuscript entitled "Proteinuria and bilirubinuria as potential risk indicators of acute kidney injury during running in outpatient settings" (medicina-936193) by Daniel Rojas-Valverde et al.

This study was conducted enrolling 29 mountain runners to explore which urinary markers could indicate AKI. The study demonstrated that 52% in AKI group runners presented significant increases in proteinuria (χ2= 0.94, p= 0.01) and 47% in bilirubinuria (χ2 30 = 0.94, p= 0.04), while No-AKI participants presented no significant urine markers increases. Thus the authors suggested the potential use of urinalysis as an accessible alternative to early identify transitional AKI until a clinical confirmation is performed.

# Generally speaking, this paper is well-written, and the topic is potentially interesting in sports medicine and nephrology.

However, some concerns should be further clarified and addressed.

[General comments]:

# The rule of abbreviation is not entirely applied in the text. (ex: AKI, BUN, sCr)

# Some typo error were noted in the main text

[Materials and Methods]

# The study design (such as prospective or retrospective, the data, and the place where the study was undergone) is lacking.

# I am not sure whether the 29 participants were gathered from a race or many races.

# The statement "There were no reported urination problems or difficulties neither before nor after the race" appeared repeated in both "2.3.2. Urine test" and "2.3.3. Urine specific gravity" sections. I suggest removing the statement in the "2.3.3. Urine specific gravity" section.

[Results]

# I would suggest the authors provide a table comparing the demographic and basic data between the "AKI group" and "No-AKI group."

# I would also encourage the authors to provide a table comparing the laboratory (blood and urinary data) between the "AKI group" and "No-AKI group."

# I do not quite understand the meaning of "(>1)" in Tables 1 and 2. The authors are required to make a more precise explanation.

[Discussion]

# As the authors' statement in the "limitation section," the total body fluid status (or "hydration" status) is crucial regarding the occurrence of AKI, but it is missing in the current study.

[Conclusions]:

# The conclusion section is a little too long. I suggest the authors move most of the content to the "Discussion" section and only left some "real conclusion" in the "Conclusion" section.

Author Response

Dear Editor and reviewers:

We have carefully considered all reviewers' recommendations of the paper ID (medicina-936193) entitled "Proteinuria and bilirubinuria as risk indicators of acute kidney injury during running in outpatient settings”. Please find enclosed our detailed answers to reviewers' queries. The authors declare that the manuscript is original and has not been considered for publication elsewhere. Additionally, the authors had approved the paper for release and are in agreement with its content.

Please find all corrections in red inside the manuscript.

Reviewer  1

R1.1. I have carefully reviewed the manuscript entitled "Proteinuria and bilirubinuria as potential risk indicators of acute kidney injury during running in outpatient settings" (medicina-936193) by Daniel Rojas-Valverde et al.

R/We really appreciate the opportunity to improve the final version of the manuscript. We considered all your suggestions and the final version is corrected accordly.

R1.2. This study was conducted enrolling 29 mountain runners to explore which urinary markers could indicate AKI. The study demonstrated that 52% in AKI group runners presented significant increases in proteinuria (χ2= 0.94, p= 0.01) and 47% in bilirubinuria (χ2 30 = 0.94, p= 0.04), while No-AKI participants presented no significant urine markers increases. Thus the authors suggested the potential use of urinalysis as an accessible alternative to early identify transitional AKI until a clinical confirmation is performed.# Generally speaking, this paper is well-written, and the topic is potentially interesting in sports medicine and nephrology.

However, some concerns should be further clarified and addressed.

R/We have considered all your suggestion to improve the final version of the manuscript. We really appreciate your time and dedication.

[General comments]:

R1.3. # The rule of abbreviation is not entirely applied in the text. (ex: AKI, BUN, sCr)

R/Thank you for pointing out this issue. It was corrected throughout the manuscript.

R1.4. # Some typo error were noted in the main text

R/We appreciate you for highlight this mistakes. All these issues were corrected.

[Materials and Methods]

R1.5. # The study design (such as prospective or retrospective, the data, and the place where the study was undergone) is lacking.

R/This information was included in the design section as recommended.

R1.6. # I am not sure whether the 29 participants were gathered from a race or many races.

R/ Thank you for the opportunity to clarify. This information was included in the participants section.

R1.7. # The statement "There were no reported urination problems or difficulties neither before nor after the race" appeared repeated in both "2.3.2. Urine test" and "2.3.3. Urine specific gravity" sections. I suggest removing the statement in the "2.3.3. Urine specific gravity" section.

R/Thank you for the suggestion. The statement was removed from 2.3.3. as recommended.

[Results]

R1.8. # I would suggest the authors provide a table comparing the demographic and basic data between the "AKI group" and "No-AKI group."

R/We included the table in results section as suggested.

R1.9. # I would also encourage the authors to provide a table comparing the laboratory (blood and urinary data) between the "AKI group" and "No-AKI group."

R/We really appreciate the opportunity to clarify. The comparison of laboratory results were presented in figure. A MANOVA was presented a by marker.

R1.10. # I do not quite understand the meaning of "(>1)" in Tables 1 and 2. The authors are required to make a more precise explanation.

R/This information regarding the result of urine dipstick was clarified in the methods section 2.3.2. Thank you for the opportunity to improve this result.

[Discussion]

R1.11. # As the authors' statement in the "limitation section," the total body fluid status (or "hydration" status) is crucial regarding the occurrence of AKI, but it is missing in the current study.

R/Thank you for the opportunity to clarify. We agree that hydration status is crucial for the analysis of AKI, that is why we present the urine specific gravity value, considering your appreciation it was additionally analyzed by group. There were no changes between pre-post test suggesting no influence of hydration status during the race.

[Conclusions]:

R1.12. # The conclusion section is a little too long. I suggest the authors move most of the content to the “Discussion” section and only left some “real conclusion” in the “Conclusion” section.

R/We agree with this recommendation, the conclusions section was limited to those real conclusions.

Reviewer 2 Report

In the manuscript, Daniel et al, studied to explore key urinary markers that could be an indicator for AKI during prolonged trail running in outpatient settings. The aim of the study is short and straight forward. The results show that there are a significant number of AKI participates developed an increase in proteinuria and bilirubinuria when compared to non-AKI participates after prolonged running. The results are clear and convincing. As a minor suggestion, the author may provide a flow chart for the study design and settings. The text is well written and has relevant references.

Author Response

Dear Editor and reviewers:

We have carefully considered all reviewers' recommendations of the paper ID (medicina-936193) entitled "Proteinuria and bilirubinuria as risk indicators of acute kidney injury during running in outpatient settings”. Please find enclosed our detailed answers to reviewers' queries. The authors declare that the manuscript is original and has not been considered for publication elsewhere. Additionally, the authors had approved the paper for release and are in agreement with its content.

Please find all corrections in red inside the manuscript.

R2.1. In the manuscript, Daniel et al, studied to explore key urinary markers that could be an indicator for AKI during prolonged trail running in outpatient settings. The aim of the study is short and straight forward. The results show that there are a significant number of AKI participates developed an increase in proteinuria and bilirubinuria when compared to non-AKI participates after prolonged running. The results are clear and convincing. As a minor suggestion, the author may provide a flow chart for the study design and settings. The text is well written and has relevant references.

R/Thank you for the opportunity to improve the final version of the manuscript. The flow chart and settings were presented in the design section.

This manuscript is a resubmission of an earlier submission. The following is a list of the peer review reports and author responses from that submission.

Round 1

Reviewer 1 Report

It is crucial and necessary to identify easily available predictors and markers of AKI to prevent, and diagnose and treat respectively, this complication, and consequently, minimize its associated morbidity and mortality and much effort has been made in this regard.

Soto K, Coelho S, Rodrigues B, et al. Cystatin C as a marker of acute kidney injury in the emergency department. Clin J Am Soc Nephrol. 2010;5(10):1745-1754. doi:10.2215/CJN.00690110

Devarajan P. Biomarkers for the early detection of acute kidney injury. Curr Opin Pediatr. 2011;23(2):194-200. doi:10.1097/MOP.0b013e328343f4dd

de Geus HR, Betjes MG, Bakker J. Biomarkers for the prediction of acute kidney injury: a narrative review on current status and future challenges. Clin Kidney J. 2012;5(2):102-108. doi:10.1093/ckj/sfs008

Lima C, Macedo E. Urinary Biochemistry in the Diagnosis of Acute Kidney Injury. Dis Markers. 2018;2018:4907024. doi:10.1155/2018/4907024

Gameiro, J., Lopes, J.A. Complete blood count in acute kidney injury prediction: a narrative review. Ann. Intensive Care 2019;9:87. https://doi.org/10.1186/s13613-019-0561-4

Gameiro J, Branco T, Lopes JA. Artificial Intelligence in Acute Kidney Injury Risk Prediction. J Clin Med. 2020;9(3):678. doi:10.3390/jcm9030678

The aim of Rojas-Valverde et al. study was “To explore which urinary markers could indicate acute kidney injury (AKI) during prolonged trail running in tropical settings” and proposes the application of it as an in-field analysis for AKI detection during prolonged trail running.

Serum blood and urine of 29 runners were analyzed before and after 35 km mountain athletic event.

The evaluated urinary parameters were: proteinuria, bilirubinuria, leukocyturia, urobilinogenuria, hematuria, nitrituria, glycosuria and ketonuria while creatinine was measured in serum.

Based on serum creatinine levels participants were distributed in two groups: AKI and non-AKI.

The main findings of this study were:

  1. a) AKI and no-AKI runners showed differences in serum creatinine levels (comparing each group values before and after running).
  2. b) AKI and no-AKI runners showed differences in serum creatinine levels after running.
  3. c) 52% of AKI runners showed an increase in proteinuria.
  4. d) 47% of AKI runners showed an increase in bilirubinuria.
  5. e) No-AKI participants showed no increase in the above-mentioned markers.

The authors concluded that data of proteinuria and bilirubinuria obtained from AKI-runners might support the potential use of urinalysis as an accessible alternative to early identify in-field AKI cases and identify transitional decreases in renal function.

OBSERVATIONS.

Question 1. If pre-race in serum creatinine values from AKI and non-Aki showed no differences, how these groups were defined?

Question 2. Approximately the half of AKI runners showed an increase in proteinuria and bilirubinuria after race, and the other half showed no such changes. That is, a half of AKI runners showed similar data that non-AKI runners, how these data support the conclusion suggested by the authors?

Question 3. Why other markers as blood urea nitrogen and uric acid were not considered for renal function test?

Question 4. Several works suggest that urinary biomarkers should be normalized to urinary creatinine levels. Why the authors did not consider the use of this normalization?

Question 5. The use of dipstick for urine analysis has negative comments, then, its use is controversial.

Park JI, Baek H, Kim BR, Jung HH (2017) Comparison of urine dipstick and albumin:creatinine ratio for chronic kidney disease screening: A population-based study. PLoS ONE 12(2): e0171106. https://doi.org/10.1371/journal.pone.0171106

Indeed, the authors mentioned that in the interpretation of dipstick data an interpretation of a third observer was necessary. So, data obtained from dipstick were compared or verified with data obtained from other methods and instruments?

Minor observations.

The conclusions section must be rewritten since contains arguments that have cited in the discussion.

In 2.3.2. Urine test section bilirubinuria parameter is not mentioned.

Please insert a space among distance, time and weight units.

Lines 22, 91. “35km event”; “1815m”

Line 96 .“15min”

Line 111. “5mL”

Line 113. “10min”

Line 189. “150mg”

Correct:

Lines 23 and 92. “1.98 oC”

Correct (grammar/spelling):

Lines 28, 171 and 239. “A total of 52%”

Line 54. “Some evidence have reported”

Line 74. “This changes in urine in characteristics are usually associate”

Line 80. “renal function have to be monitor in endurance sports”

Line 145. “There were an large differences”

Line 154. “leucocyturia”, “urobilonogenuria”, “glucosuria”.

Line 168. Acute kidney injury has been defined in the “Introduction” section.

Line 173. “No-AKI participants presented no urine markers significant increases”

Line 176. “In sports, precisely in endurance sports the mechanisms”

Line 178. “In strenuous exercise high physical load during prolonged periods have an essential role”

Line 179. “Relative common rise in”

Line 181. “damage and disintegration of muscle fibers is a common consequence”

Line 190. “this conditions is known as proteinuria could be developed”

Line 198. “catecholamines are release by the suprarenal glandules”

Line 206. The increase in bilirubin foun in this“

Line 238. “This study found Pre vs post-race differences”

Line 241. “a decrease in renal and kidney function”

Please increase the size of text and numbers in graph of Figure 1.

Line 67. Please clarify the following phrase “This diagnosis markers and criteria allows not only earlier identification of AKI but also a diagnosis even in the absence of subsequent kidney dysfunction”

“The extracted variable was” ???

Reviewer 2 Report

This paper entitled “Proteinuria and bilirubinuria as risk indicator of acute kidney injury during running in tropical setting” demonstrated that proteinuria and bilirubinuria might be able to predict the occurrence of AKI after running in tropical circumstances. This paper is of very interest for better understanding the mechanism of exercise-induced AKI and seems to be important for clinicians. This paper looks interesting, but there are some methodological issues and problems to be resolved.

Major concern,

  1. Long-distance run induce severe hydration which, in turn, leads to increase in the concentration of protein in urine. Authors used only dipstick to analyze urine protein; thus, they can’t exclude the possibility that dehydration-induced change in concentration of urine protein affect the data. Please demonstrate the estimated urine protein (g/gCre) which seems to be definitely necessary for this research.
  2. Authors showed that proteinuria or bilirubinuria-positive population was increased due to long-distance run in AKI group. Meanwhile, there are also some proteinuria or bilirubinuria-positive individuals in non-AKI population; but, there is no statistical differences between pre and after running. However, the differences in sample size between two groups probably disturb the accurate assessment about their data. Authors should calculate the adequate sample size for this study and if necessary please add samples. Also, authors should explain why there is no statistical difference in occurrence of proteinuria even if 33% of non-AKI individuals showed proteinuria after running.
  3. Authors should show the change in body weight (dBW) between pre and after long-distance running which can provide some information about dehydration. In this research, running-induced dehydration is highly likely to affect the levels of proteinuria measured by dipstick. Pease re-analyze the data with concerning dBW.
  4. Authors should show what kind of protein were increased or secreted in urine after run. That would be of very interest for better understanding the difference between AKI group and non-AKI group.
  5. Authors mentioned that “Vasoconstriction of glomerular arteriole is also provoked by catecholamines resulting in increased glomerular filtration pressure” in line 200-201. It looks weird because it is generally accepted that vasoconstriction of glomerular arteriole should reduce the blood flow into glomeruli, leading to the decrease in glomerular filtration pressure. Please explain deeply and accurately.

Reviewer 3 Report

In this original article, Rojas-Valverde et al. are presenting an analysis of acute kidney injury (AKI) in endurance runners taking part to competitions in tropical settings.

The study has been executed on 29 experienced runners, evaluated before and after a 35km run with blood and urine analysis.

The study population has been divided into No-AKI and AKI, according to the post-race levels of serum creatinine, matching the criteria for AKI diagnosis. Urinalysis was conducted in parallel, leading to the evidence that proteinuria and bilirubinuria are the only analytes that show a significant difference between the pre and post evaluation in the AKI group but not in the No-AKI. Therefore, the Authors suggest a possible role for proteinuria and bilirubinuria in the early detection of AKI in runners.

Personally, I think the work is quite preliminary (as also stated by the Authors themselves, since the cohort is quite small). I’ll express here my concerns:

  1. To my opinion the text needs quite a bit of rephrasing since it is difficult to read and it has quite a bit of spelling and grammar errors.
  2. Anyway, my main concern resides with the data themselves: how would the Authors argument on the fact that the statistical significance in bilirubinuria in AKI is 0.039, while in the No-AKI is 0.063, somehow both very close to the cut-off. This to me might just suggests a trend due to the small cohort studied.
  3. Is there any variable that is able to predict who is going to develop AKI before testing the post-race serum creatinine or the post-race urinalysis analytes?